# Intermittent Fasting and Reduction of Inflammatory Response in a Patient with Ulcerative Colitis

**DOI:** 10.3390/medicina59081453

**Published:** 2023-08-11

**Authors:** Ángel Roco-Videla, Claudio Villota-Arcos, Carolina Pino-Astorga, Daniela Mendoza-Puga, Mauricio Bittner-Ortega, Tatiana Corbeaux-Ascui

**Affiliations:** 1Facultad de Salud y Ciencias Sociales, Universidad de las Américas, Providencias, Santiago 7500975, Chile; 2Escuela de Nutrición y Dietética, Facultad de Ciencias de la Salud, Universidad Bernardo O’Higgins, Santiago 8370993, Chile; carolina.pino@ubo.cl; 3Laboratorio de Microbiologia y Biotecnologia Oral, Facultad de Ciencias de la Vida, Universidad Andres Bello, Echaurren 237, Piso 6, Santiago 8370133, Chile; mbittner@unab.cl; 4Integramédica, Parte de BUPA, Director Médico ITR-IBB-ITA, Cerro Colorado 5240, Las Condes, Santiago 7560995, Chile; daniela.mendoza@integramedica.cl; 5Fundación Oncoloop, Av. Antonio Varas 710, Providencia, Santiago 7500967, Chile; tatiana@oncoloop.cl

**Keywords:** ulcerative colitis, intermittent fasting, CRP, fecal calprotectin, inflammation

## Abstract

Ulcerative colitis is an inflammatory disease that affects the colon, generating a crisis period associated with diarrhea and ulcerations. Stress plays a pivotal role in modulating the inflammatory response and aggravating progression. Different studies have shown that fasting reduces inflammation markers, and intermittent fasting decreases inflammatory markers such as IL-2, IL-6, and RCP. Goal: To evaluate the impact of intermittent fasting on a patient diagnosed with ulcerative colitis. A female patient underwent intermittent fasting (10/14) for eight weeks. Clinical tests were performed for blood count, RCP, biochemical profile, glycemia, and T4/TSH levels. Fecal calprotectin was determined. Clinical exams were assessed before and after intermittent fasting. Inflammation markers, such as CRP and calprotectin, were significantly reduced after eight weeks of intermittent fasting. The patient reported feeling better and was seizure-free during the following months when she continued fasting intermittently. Intermittent fasting allowed for a reduction in inflammation markers.

## 1. Introduction

Ulcerative colitis is an inflammatory disease that affects the colon, generating a crisis period associated with diarrhea and ulcerations. Patients have symptomatic periods (flare-ups) and asymptomatic periods (remission) [1,2]. Therapeutic management includes anti-inflammatory drugs, such as corticosteroids and 5-aminosalicylates, which have limited efficacy and cannot be used for long periods. When immune system inhibitors, such as azathioprine, mercaptopurine, and methotrexate, are used, patients experience several side effects, causing liver inflammation, nausea, and vomiting. Other drugs, such as natalizumab, vedolizumab, infliximab, adalimumab, and certolizumab, inhibit the function of immunoregulatory molecules such as IL-6 and TNF-α [3,4]. Symptoms are treated with antidiarrheals, pain relievers, vitamins, and supplements. However, the benefits of these treatments are temporary, and each is associated with risks inherent to their procedure and side effects [3,4]. Different diets, such as Atkins, Zone, Weight Watchers, Ornish, ketogenic diet, and intermittent fasting (IF), have been tested as an alternative for preventing chronic non-communicable diseases [5]. 

In a study carried out in mice, the effects of intermittent fasting were analyzed in the short and long term (2 weeks and 20 weeks). Results showed that both periods of time led to a significant impact on the body weight and adipose tissue content of mice, and changes were observed in the bile acid profile of feces. Fecal metabolome analysis identified several metabolites that were enriched in the intermittent fasting groups, including glucose, kynurenic acid, inosine, and 3,4-dihydroxyphenylacetic acid (3,4-DHPA). Kynurenine reduces the differentiation of T cells into highly inflammatory Th17 cells. Inosine also has anti-inflammatory effects on human monocytes and neutrophils. 3,4-DHPA can inhibit the secretion of pro-inflammatory cytokines in lipopolysaccharide-stimulated peripheral macrophages [6,7]. Differences in gut microbiota composition were also found between intermittent fasting and ad libitum feeding groups. These findings suggest that intermittent fasting might have an impact on the metabolism and composition of the gut microbiota [7]. It has been reported that during Ramadan, when the population eats in a restricted manner between 12 and 18 h a day, weight loss occurs, as well as a regulation of lipid and glucose levels. The latter is accompanied by a decrease in the levels of inflammation markers, such as IL-1 and IL-6, CRP, and TNF-α [8,9]. Furthermore, intermittent fasting in combination with physical exercise has been shown to improve LDL and HDL concentrations in plasma. 

Thus, fasting reduces the risk of coronary disease [10]. Another study showed a statistically significant decrease in systolic and diastolic blood pressure along with decreased body weight in women who followed IF for six weeks [11]. Intermittent fasting has also been shown to have beneficial effects on blood glucose levels, with a very positive result in patients with C-reactive protein levels greater than 1 mg/L, which provides evidence of its effectiveness in reducing inflammatory conditions [12]. 

In an investigation using a mouse model, intermittent fasting was found to be able to reduce intestinal inflammation, decreasing CD4^+^ T cells in mesenteric lymph nodes. CD4^+^ CD25^+^ regulatory T cells (Treg) play a pivotal role in the regulation of immune responses, thus preventing autoimmunity and inflammatory responses. Treg cells modulate CD8^+^ T cell differentiation and the effector function by regulating IL-2 homeostasis [13]. In this regard, a report has shown an increase in CD4^+^ CD25^+^ regulatory T cells in the mesenteric lymph nodes and a decrease in the infiltration of leukocytes and macrophages around the base of the crypts in the colon. These results indicate that intermittent fasting may have modulatory effects on the immune system and reduce intestinal inflammation [14].

There are few studies that have assessed the mechanisms involved in anti-inflammatory responses induced by intermittent fasting. A short-term fasting study showed decreased monocyte metabolic and inflammatory activity. This study showed that a reduction in the inflammatory response was associated with the regulation of the number of peripheral monocytes, which were dependent on glucose and protein intake. A reduction in the number of monocytes was mediated by the activation of AMPK and PPARα pathways [15]. Another study in animal models has shown that IF reduced the composition of T cells in the lamina propria of the intestine with a reduction in IL-17-producing T cells and an increase in the number of Tregs. These effects also increased the richness of gut bacteria and activated microbial metabolic pathways that modulated systemic immune responses, thus concluding that intermittent fasting has potent immunomodulatory effects that are at least partially mediated by the gut microbiome [16]. 

Moreover, several observational studies have shown that reducing food intake improves digestive symptoms. However, few bodies of evidence have shown how intermittent fasting can modify symptoms in patients with ulcerative colitis and how this affects markers of inflammation such as CRP and fecal calprotectin [17]. Most human studies have been carried out on the Muslim population during Ramadan. For a month, people fasted during light periods. Studies conducted on this population have shown conflicting data. An additional study on 60 patients with IBD (43 UC, 17 CD) in remission (with no history of infection, perforation, or other comorbidities) showed no significant risks to patients with mild and uncomplicated IBD [18]. Patients with UC also showed a significant decrease in the clinical colitis activity index [19]. 

A recent study evaluated CRP and fecal calprotectin levels before and after Ramadan in a cohort of 80 patients (60 UC and 20 CD) with no comorbidities. In detail, some of the published data showed that obese subjects, after three weeks of intermittent fasting, had reduced CRP levels of 8 to 5 mg/dL [17]. In another study, fecal CRP and fecal calprotectin levels were observed to be reduced (PCR: 5.3 v/s 5.0 mg/dL) (fecal calprotectin: 163 v/s 218 mg/kg) in patients with ulcerative colitis and Crohn’s disease that had undergone one month of intermittent fasting [18]. Similar results were observed in a group of younger subjects with a significant reduction in CRP from 5 to 2.5 mg/dL after 30 days of intermittent fasting [20]. The following section describes a case report of a 42-year-old female patient with ulcerative colitis who underwent intermittent fasting for two months. The results show a reduction in fecal, calprotectin, and PCR inflammation markers.

## 2. Case Report

A 42-year-old female patient was diagnosed in 2010 with ulcerative colitis in the remission period. The patient lives in the metropolitan region of Santiago, Chile. The patient was contacted through the Carlos Quintana Foundation for Crohn’s and Ulcerative Colitis. The patient had a daily consumption of mesalazine and aspirin. No allergies were declared. The patient accepted her involvement by signing the informed consent to participate in a pilot study to determine the adherence of patients with inflammatory bowel disease to intermittent fasting (add according to Helsinki regulations, etc.). Figure 1 shows the study evaluations and interventions flow. A group of psychologists evaluated the patient by applying the 5-factor test [21,22]. This test allowed the participants to be classified into adherence profiles according to their personality profiles. The patient was classified in the group with the highest expected adherence. The patient’s general condition was evaluated by means of serological tests and the detection of calprotectin as a marker of intestinal inflammation. Table 1 summarizes the most important findings of the initial and final examination. Briefly, before starting intermittent fasting, the patient showed the following values: hematocrit: 40.4%; hemoglobin 13.8 g/dL; total leukocytes: 7.4 × 10^3^/mL; Calprotectin: 139 mg/Kg and CRP: 3.64 mg/L.

A nutriologist evaluated the patient by authorizing her participation in the study. The intervention consisted of eight weeks of intermittent fasting for 10/14. The day was divided into 10 h for food and 14 h for fasting. The adherence to fasting was determined by a weekly self-report that the patient answered. Patient-reported adherence to intermittent fasting was greater than 96%. The nutritional recommendations given to the patient are shown in Table 1. After eight weeks of intervention, the serological markers and fecal calprotectin were evaluated again. The main findings are shown in Table 2. Briefly, after eight weeks of intermittent fasting, the patient showed the following values: hematocrit: 42.1%; hemoglobin 13.8 g/dL; total leukocytes: 8.6 × 10^3^/mL; Calprotectin: 51 mg/Kg and CRP: 1.57 mg/L. The patient reported decreased inflammation and the absence of other gastrointestinal symptoms. Currently, the patient continues intermittent fasting at 10/14 and is controlled by medical staff.

## 3. Discussion

Inflammatory bowel disease (IBD) is described as a chronic inflammation of the gastrointestinal tract, leading to tissue damage, malabsorption, and systemic complications. Ulcerative colitis (UC) is determined by superficial and continuous colon ulceration with rectal complications [23]. IBD patients avoid dietary ingestion to minimize gastrointestinal complications. Psychological stress can predispose patients to disease episodes [24]. Inflammatory bowel disease is also associated with dysbiosis [25]. UC represents a risk for mental health problems, including depression and anxiety-like behaviors [22,25]. Dysbiosis plays a pivotal role in the deregulation of the Brain-GIT axe [25,26]. IF is a group of periodic energy restriction dietary patterns, including alternate-day fasting (ADF), time-restricted fasting (TRF), and intermittent energy restriction (IER) [27]. 

Previous research has reported that IF has beneficial effects on the compositions of gut microbes in animal models and human trials [28]. 

The results published by Hu et al. have shown that intermittent fasting generates changes in the microbiota by increasing the number of bacteroides and parabacteroides, which are known to be involved in beneficial effects on health [29]. In addition, enrichment of GABA-producing P. distasonis and B. thetaiotaomicron was observed. This neurotransmitter has been associated with an anti-inflammatory response [30].

Khan’s et al. previous results have shown that IF increases the number of Actinobacteria [31]. Bifidobacteria, which belong to this bacterial phylum, have been described for their importance and their anti-inflammatory response [32]. Supplementation with bifidobacteria has been shown to reduce TNF-α expression in mice, leading to a reduction in inflammatory responses [33]. Bifidobacteria release acetate, which is used by other fermenting bacteria to synthesize butyrate and propionate. These short-chain fatty acids have been shown to have the ability to reduce inflammatory response by promoting and regulating the regulatory T population of the colon [34]. The activity of dendritic cells and T lymphocytes was modulated [35].

Publications have shown, both in animal models and in clinical trials, that intermittent fasting can lead to symptom reduction in young IBD patients [10,11,12,15]. However, few studies have looked at the impact of intermittent fasting in patients with ulcerative colitis and how this affects inflammatory markers, such as the C-reactive protein and fecal calprotectin [28]. Studies have reported that Ramadan fasts are associated with significantly lower concentrations of inflammatory markers, such as CRP, IL-6, and TNF- α [19,20]. This small number of participants limits the results in relation to Ramadan; however, they showed that, in the case of UC, there is a duality of effects. It seems to be more beneficial in younger people when compared to an older population [14,18,36]. 

In studies in which the CRP and fecal Calprotectin levels have been determined, contradictory results have been shown. Negm et al. analyzed CRP and fecal calprotectin levels in patients with ulcerative colitis. The results showed that after 30 days of intermittent fasting, the mean CRP value increased, on average, from 1.17 to 1.85 mg/L for 60 patients (UC). Fecal calprotectin also increased, on average, from 176 to 218 mg/Kg. Despite the fact that their results were not shown to be statistically significant, it was reported that many patients had to drop out due to increased symptoms [18]. 

In our clinical case, we observed a significant reduction in CRP from 3.64 to 1.57 mg/L and in fecal calprotectin from 139 to 51 mg/Kg. The patient always showed a reduction in symptoms while performing intermittent fasting for two months with very high adherence (which was determined by weekly self-reports).

El Mountassir et al. observed that fasting was tolerated in 94% of cases. No negative symptoms associated with fasting were reported [37]. Aksunger et al. carried out a study and showed that, after 30 days of intermittent fasting, CRP levels were significantly reduced from 5 to 2.5 mg/L. The authors did not determine the levels of fecal calprotectin [20]. Unalacak et al. showed similar results in obese patients, in which CRP levels were shown to decrease from 2.83 to 2.7 mg/L after 30 days of intermittent fasting [38]. Widhani et al. also showed a significant reduction in CRP in HIV patients during two weeks of intermittent fasting [39].

Interestingly, this case report is one of the few interventional studies performed in humans. It shows how a 42-year-old female patient with ulcerative colitis, in the remission period and without comorbidities, underwent intermittent fasting (10/14) for two months. Clinical markers, including complete blood count, lipid profile, liver profile, C-reactive protein, and fecal calprotectin, showed a reduction in inflammatory processes. These clinical findings correlate with a reduction in the patient’s symptoms. The patient has continued to perform intermittent fasting and is regularly monitored by a medical doctor.

## 4. Conclusions

In the present study, a 42-year-old patient with UC in remission underwent IF (10/14) for eight weeks. The patient had no problems adhering to intermittent fasting. After two months of fasting, no appearance of gastrointestinal symptoms was reported. A significant reduction in CRP and fecal calprotectin was observed, as previously published by other authors. There was no evidence of an alteration in blood markers as well as alteration in the lipid and hepatic profile. 

Although this research has limitations, such as the follow-up time that restricted the ability to observe long-term changes in health markers, lipid, and liver profiles, several aspects are presented that can be considered novel. One of these precisely includes the evaluation of multiple clinical variables with a comparison before and after the period of intermittent fasting, accompanied by the intake recommendations based on easily accessible foods that ensured sources of vitamins and minerals, as well as elements with anti-inflammatory properties. Another contribution of this research is related to the fact that it is a study in a specific population of which there is little information regarding the effectiveness of intermittent fasting, such as Chilean women. The Latin American demographic group, especially the Chilean population, has not been widely studied in terms of the effectiveness of intermittent fasting; therefore, this research is a relevant contribution as a source of background information regarding this demographic group.

In general terms, it can be noted that these findings suggest that intermittent fasting may be a safe and viable strategy to improve health in Chilean women since no adverse effects on liver function or lipid metabolism were found. However, further studies are needed. In this regard, additional long-term randomized controlled studies are required to confirm these findings and assess the effects of intermittent fasting diets in different populations and in combination with other interventions and dietary recommendations. Research is also required to understand the underlying mechanisms of how the intermittent fasting diet affects health and lipid and liver profiles in relation to the diet of the Latin American population and especially the Chilean population.

## Figures and Tables

**Figure 1 medicina-59-01453-f001:**
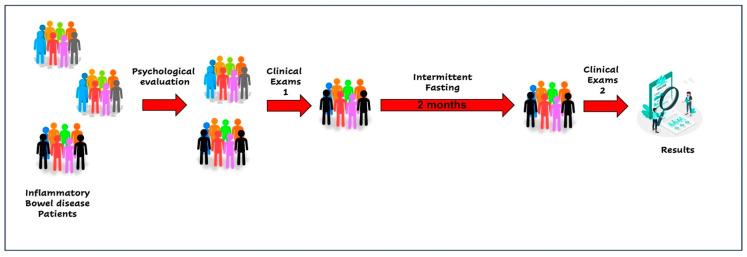
Intermittent fasting intervention diagram.

**Table 1 medicina-59-01453-t001:** Results of clinical evaluation before (Pre-T) and after two months (post-T) of intermittent fasting.

**Blood Count**	**Pre-T**	**Post-T**	
Hematocrit	40.4	42.1	%
Hemoglobin	13.8	13.8	gr/dL
Total Leukocytes	7.4	8.6	10^3^/mL
Platelets	348	359	10^3^/mL
HSV	11	10	mm/Hr
**Coagulation**	**Pre-T**	**Post-T**	
INR	1	1	
**Lipidic profile**	**Pre-T**	**Post-T**	
Total Cholesterol	174	169	mg/dL
HDL	61	61	mg/dL
LDL	90	87	mg/dL
TG	112	103	mg/dL
**Inflammation**	**Pre-T**	**Post-T**	
Calprotectin	139	51	mg/Kg
CRP	3.64	1.57	mg/L
**Glucose**	**Pre-T**	**Post-T**	
Basal	90	95	mg/dL
OGTT	135	119	mg/dL
**Liver profile**	**Pre-T**	**Post-T**	
GGT	10	13	U/L
AP	96	91	U/L
GOT/AST	18	17	U/L
GPT/ALT	19	16	U/L
Bilirubin	0.42	0.43	mg/dL
**Thyroid**	**Pre-T**	**Post-T**	
TSH	3.7	4.53	uUI/mL
Free T4	1.01	1.05	ng/dL

**Table 2 medicina-59-01453-t002:** Meal plan recommendations for intermittent fasting.

**Nutritional Recommendations for Intermittent Fasting 10/14**	
Feeding time 10 h	
intermittent Fasting 14 h	
Fasting	
During fasting you can drink water and infusions such as tea or chamomile sugarless/sweetener less	
During fasting you can´t eat neither drink liquid with sugar or sweetener, coffee, natural juice, or powder juice	
**Feeding time**	
**Sources**	**Foods**
Vit E	Nuts such as almonds, hazelnuts, peanuts Sunflower oil, corn, Spinach, broccoli, kiwi, tomato, mango.
Vit C	Paprika Orange Grapefruit, Kiwi, Broccoli Strawberries, Tomato, Sprouts (cabbage, cauliflower, Brussels).
Vit D	Cod oil, Egg, Liver, Salmon, tuna, Mushroom (exposed to UV light), Vegetable enriched drinks
Vit B-complex	Liver, Spinach, Asparagus, Cabbages, Legumes, Avocado, Chicken, Salmon, Tuna Clams, Trout, Turkey, Beef, fortified vegetable drinks.
Antiox	Grape, Tomato, Cocoa, Blueberries, Strawberries, Green tea, Carrot, Maqui, Calafate, Murtilla.
Iron	Fortified cereals, Legumes, Seafood, Dark green vegetables, Fortified vegetable drinks, Food of animal origin.
Calcium	Dairy and derived products, these products can be whole, semi-skimmed or skimmed according to tolerance. It can also be lactose-free according to tolerance. Cereals like amaranth, Legumes such as beans, chickpeas, tofu, Nuts such as almonds and sesame seeds, Calcium-fortified vegetable drinks.
Magnesium	Pumpkin seed, Quinoa, Spinach, Walnuts, almonds, pine nuts, Beans.
Natural anti-inflammatories	A good consumption of fruits and vegetables is recommended, avoiding highly processed foods and refined carbohydrates, cakes, cookies, etc. Foods with omega 3 content: Fatty fish such as salmon, tuna, horse mackerel, etc, walnuts, flaxseed, flaxseed oil, olive oil, pumpkin seeds. Promote the consumption of whole foods such as rice, whole wheat bread and noodles, amaranth, quinoa.Spices such as turmeric, ginger, rosemary (according to tolerance).

## Data Availability

The data presented in this study are available on request from the corresponding author.

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
