# Peer review of "Intermittent Fasting and Reduction of Inflammatory Response in a Patient with Ulcerative Colitis"

_medicina, 2023, doi:10.3390/medicina59081453_

Round 1

Reviewer 1 Report

The manuscript by Ángel Roco-Videla et al. repoted a case about the intermittent fasting and reduction of inflammatory response in a patient with ulcerative colitis. This case report is of interest to the readers and I have the following suggestions:

1, The English of this manuscript must be further improved. So many grammatical errors. Extensive editing of English language required.

2, The introduction should be split into three or two sections. The potential mechnisms underlying the intermittent fasting and reduction of inflammatory responses should be discussed. 

3, The potential effects of intermittent fasting on the gut microbiota and the reduction of inflammatory response should be discussed. Gut microbiota may be changed in response to intermittent fasting and thus it may affect the inflammatory responses in the gut. 

The English of this manuscript must be further improved. So many grammatical errors. Extensive editing of English language required.

Author Response

Request 1:

The English of this manuscript must be further improved. So many grammatical errors. Extensive editing of English language required.

Action realized:

Review of a native speaker was requested regarding the use of language.

Request 2:

The introduction should be split into three or two sections. The potential mechanisms underlying the intermittent fasting and reduction of inflammatory responses should be discussed. 

Action realized:

The introduction was split into three sections. The mechanisms the intermittent fasting and reduction of inflammatory responses were included in introduction section.

Request 3:

The potential effects of intermittent fasting on the gut microbiota and the reduction of inflammatory response should be discussed. Gut microbiota may be changed in response to intermittent fasting and thus it may affect the inflammatory responses in the gut. 

Action realized:

The potential effects of intermittent fasting on the gut microbiota and inflammatory response were included in discussion section.

Reviewer 2 Report

Roco-Videla et al. present an interesting case report about the role of intermittent fasting in ulcerative colitis and its consequences in terms of inflammatory markers.

They report the experience of a 42-year-old female patient with ulcerative colitis, in remission period and without comorbidities, undergoing intermittent fasting for two months.  Clinical markers and fecal calprotectin, showed a reduction in inflammatory processes. Patients' symptoms followed the same variation. 

The results are promising but I think the manuscript could be improved reporting the limits of a case report and giving the future prospective of a case series or prospective population enrolled with the same study timeline and criteria. Furthermore, the results are expected and in line with previous published studies. I suggest to enlight novelties or new insights driven by this single case experience, in order to catch more attention and give interest in future large studies. 

The English is good. Minor editing is required (e.g. line 154 the sentence has no conclusion...by a medical?"). 

Author Response

Request:

The results are promising but I think the manuscript could be improved reporting the limits of a case report and giving the future prospective of a case series or prospective population enrolled with the same study timeline and criteria. Furthermore, the results are expected and in line with previous published studies. I suggest to enlight novelties or new insights driven by this single case experience, in order to catch more attention and give interest in future large studies.

Action realized:

Comments on the contribution and novelty that this research implied and possible future research based on the results obtained to make the publication more striking were included in the conclusions.

Request:

The English is good. Minor editing is required (e.g. line 154 the sentence has no conclusion...by a medical?").

Action realized:

Review of a native speaker was requested regarding the use of language.

Round 2

Reviewer 1 Report

The authors have revised the manuscript accordingly. It can be considered for publication.